# The Influence of Power on Leisure: Implications for Inclusive Leisure Services

**DOI:** 10.3390/ijerph18052220

**Published:** 2021-02-24

**Authors:** Francisco Javier Lopez Frias, John Dattilo

**Affiliations:** 1Kinesiology Department and Rock Ethics Institute, Pennsylvania State University, University Park, PA 16802, USA; 2Recreation, Park and Tourism Management, Pennsylvania State University, University Park, PA 16802, USA; jxd8@psu.edu

**Keywords:** inclusion, leisure, power, Foucault

## Abstract

Many people experience domination as they encounter oppression and marginalization because of power differentials limiting their leisure. We rely on Foucault for guidance to examine connections between power and opportunities for people to be included in leisure and recognize that, like Foucault, we experience privilege. Considering such privilege, we explore power and people connections, scrutinize ways power influences leisure, and examine methods to promote or resist power to increase leisure. Drawing on the analysis of power and leisure, we examine how discourse influences leisure and identify ways to facilitate inclusive leisure. We consider these aspects via Allen’s (1998) modalities of power-over, power-to, and power-with. Analyzing these modalities, we address barriers to leisure associated with power, strategies people use to engage in resistance through leisure, and ways inclusive leisure might occur. We conclude that each person can make positive contributions and offer inclusive leisure.

## 1. Introduction

According to Foucault, philosophy is a way of reflecting on people’s relationship with truth. In his words, philosophy is “the movement by which…one detaches oneself from what is accepted as true and seeks other rules…The displacement and transformation of frameworks of thinking” [1] (p. 327). Thus, a fundamental function of philosophy is to call into “question domination at every level and in every form in which it exists, whether political, economic, sexual, [or] institutional” [1] (pp. 300–301). As von Wright posits, the humanities in general, and philosophy in particular, involve the study of humans as cultural beings [2] (p. 156). Humans create symbolic or discursive frameworks to make sense of the world, find a place in it, and develop ways to collaborate in order to survive and thrive. This is why Aristotle identifies *logos* (which translates to English as “reason” or “language”) as humans’ key defining element [3], and Heidegger famously said that “language (*logos*) is the house of Being” [4] (p. 313). Similarly, Ortega y Gasset explains, whereas biological forces strongly influence animal behavior, human behavior extends beyond biology because humans have history. That is to say, human behavior results from interactions between individuals and their cultures. Cultures comprise values, ideas, norms, and social practices [5]. Understanding these phenomena is the main goal of philosophy, for it centers on ideas, values, norms, and social practices on which individuals rely to make sense of their lives and live together. “Understanding” here means to reflect on “aims and purposes of an agent, the meaning of a sign or symbol, and the significance of a social institution or religious rite” [6] (p. 6) and differs from ways in which empirical scientists acquire knowledge (which von Wright refers to as “explanation” [6]). In this paper, we analyze philosophical theories and concepts to reflect on connections between leisure and discursive frameworks and power, which itself is another cultural object resulting from human actions and beliefs.

Much literature on inclusive leisure focuses on people with a specific characteristic, such as disability, immigration status, age, income, residence, sexual orientation, religion, and ethnicity or race (see [7,8]). Continued examinations of specific groups contribute to understanding ways in which group members often encounter suppression and promote solidarity within leisure enclaves. When considering these distinctive groups, “It has been no easy thing to go public, to fight against cultural odds, against insults and scorn, often against physical violence” [9] (p. 144). However, there are “distinct but interrelated lines of inquiry that bring attention to the issues of power, inequality, and privilege and espouse change for marginalized populations” [10] (p. 3).

Each group of people encounters oppression differently depending on nuanced histories of inequality; each with their embeddedness, performance within, and structures associated with power through leisure. Consequently, scholars may feel a need to study such groups individually. However, Gitlin [9] warns that efforts creating distinct cultures to address unique group concerns promote fragmentation, resulting in knowledge silos and power hoarding [11]. Blanton and Pugach [12] conclude that research on education for inclusion is increasingly available, yet it is fragmented primarily because models have yet to be developed explaining how best to prepare professionals to serve all participants well. Aitchison [13] advocates inclusive and discursive exchanges between areas of study by avoiding social exclusionary discourses to develop comprehensive theory, policy, and practice.

To be included is to be a part of a group, often involving the presence of someone in a community [14]. According to Carruthers and Hood [15], inclusion infers that all people have the right to be a part of their communities. Focusing on inclusion of all people, rather than explicitly on accommodating specific groups, avoids fracturing efforts for equity by examining commonalities across marginalized groups to discover mutual challenges and experiences [16]. Inclusive services involve meeting needs of participants who vary based on their backgrounds, cultures, and life experiences, as well as innate and acquired characteristics that both can enrich and limit a person [17]. Inclusive leisure services, thus, aim to provide “necessary support and flexibility to provide opportunities for people having diverse backgrounds, cultures, life experiences, as well as innate and acquired characteristics, to have choices on ways to experience leisure that contribute to their overall happiness and ability to flourish” [18] (p. 72). Therefore, one of the main goals of inclusive leisure contexts is to create awareness of similarities and limit concentration on differences [19]. “To achieve inclusive leisure, we counter oppression by creating accommodating environments as we consider unique needs of each participant; we foster reciprocity among participants to encourage feelings of acceptance and being welcome” [20] (p. 181).

In this paper, we use a Foucauldian perspective to analyze a phenomenon that influences people’s opportunities to experience leisure. This phenomenon is power. Although power adopts different configurations depending on the community and situation, we can understand ways in which it connects to leisure that are applicable to all communities. That is to say, we can provide an examination of power that centers on how power influences opportunities to experience leisure and ways in which communities and providers, generally speaking, can account for power. To do so, we consider Allen’s [21] analyses of power in feminist theory and address three questions: (1) How does power-over create barriers to leisure? (2) What are ways to promote power-to that facilitate access to leisure? (3) How do people collaborate to exert power-with? In answering these questions, we identify considerations for developing inclusive leisure.

## 2. The Pervasiveness of Power

According to Rojek [22], scholars often contrast leisure to other activities, such as work, and ignore power. Those who hold this notion of leisure, in Rojek’s view, abstract leisure from contextual conditions, failing to understand fully its nature. Miller [23] posits that Foucault’s account runs counter to interpretations of leisure that connect it to individual consciousness and values (e.g., leisure as a commodity or entertainment), as well as those who link leisure to modern civilization and its institutions, especially the market and the state (see [24]). Foucault’s analyses of power move discussions on leisure to a higher level of generality, namely that of discourse and cultural institutions. Individual consciousness, personal values, states, institutions, and conceptualizations of leisure vary with time and space; they depend on discourse. Foucault describes discourse, or more precisely, discursive formation, as follows: “Whenever one can describe, between a number of statements, such a system of dispersion, whenever, between objects, types of statements, concepts, or thematic choices, one can define a regularity (an order, correlations, positions and functionings, transformations), we will say, for the sake of convenience, that we are dealing with a discursive formation” [25] (p. 38). Discourse includes statements that frame ways to articulate knowledge [26]. For Foucault, a statement is “a function of existence that properly belongs to signs and on the basis of which one may then decide… whether or not they ‘make sense’… and what sort of act is carried out by their formulation.” [25] (p. 87) Thus, statements make knowledge possible by delimiting what people can say and who can speak, as well as determining in what sense and with what authority one can do it [27].

Since discourse exerts power on people by delimiting what they can say, know, and do [28], power is not something that people who experience privilege possess; rather it is a result of human actions. In Foucault’s words, “power does not exist… power only exists when it is put into action” [29] (p. 788). This is why, for Foucault, discourse is a practice or active engagement. Power is a relational phenomenon that results from action. By acting, people give meaning to the world and organize social institutions, forming discursive fields. Bourdieu [30] concludes that discursive spaces contain meanings communicated via discourse involving production and replication of power relations within institutional spaces as a way to create domination.

Often, scholars take power to refer to situations in which a person or group relies on authority to get a person or group to do something they would not do otherwise, at times violating human rights [31]. For instance, Weber defines power as “the probability that one actor within a social relationship will be in a position to carry out his [sic] own will despite resistance” [32] (p. 53). Similarly, Dahl connects power to conflict of interests in which people impose their will on others: “A has power over B to the extent that he [sic] can get B to do something that B would not otherwise do” [33] (pp. 202–203). Influenced by these two definitions, many discussions associated with power focus on political institutions and figures since their function is to exert authority to govern; such views of power conceive of society in binary terms [34]. For instance, Marxist and feminist scholars identify two opposing modalities of power (i.e., domination and resistance), and divide society into oppressed (i.e., workers and women) and oppressors (i.e., people who are wealthy and male) [21].

As Sharpe explains, Foucault rejects “conception of power as binary” [34] (p. 916), providing a broader notion of how people collectively create and shape their identity through discursive spaces and processes, such as what constitutes knowledge, what people can say, who speaks, and who decides [35]. According to Foucault [29], power results from people’s actions in social networks; it does not refer to authoritarian or institutional relationships, but to relationships between people broadly speaking. Because power results from human action, everyone can exert it. The dichotomous distinction between powerful and powerless people is unsound. Power occurs in any situation in which people make choices, not only in situations in which some people impose their authority (i.e., the power they possess) on others: “[t]he exercise of power is not violence; nor is it consent which, implicitly, is renewable. It is a total structure of actions brought to bear upon possible actions; it incites, it induces, it seduces, it makes easier or more difficult… it is nevertheless always a way of acting upon an acting subject or acting subjects by virtue of their acting or being capable of action.” [29] (p. 789) To analyze leisure, following Foucault’s approach to sex as a discursive object, one must account for “the fact that is spoken about, to discover who does the speaking, the positions and viewpoints from which they speak, the institutions which prompt people to speak about it, and which store and distribute the things that are said. What is at issue is the way in which it is ‘put into discourse’” [36] (p. 11).

A Foucaldian analysis of leisure requires examination of ways people talk about, act upon, and reflect on leisure experiences. Some discursive fields, for instance, might result from people experiencing freedom and well-being while engaged in leisure. However, in other discourse fields, leisure ties with experiences of exclusion, suffering, and discrimination. Thus, ways in which people talk about, think of, and act upon leisure radically vary, creating different types of discourses and power relations.

## 3. Power in Leisure

Mulvey et al. [37] recognize power is a part of every encounter, often unnoticed by those privileged in situations, including situations involving leisure [38]. As Foucault observes, though people engage in various cultural practices and believe ramifications of actions are obvious, “[power is] far from self-evident and [is] the culmination of a very long history.” [39] (p. 97). Since power results from action, it consistently shapes leisure and associated perceptions, with power relationships often being unequitable and resulting in dominant people accessing and controlling leisure [40]. Scholars and practitioners better understand the leisure phenomenon and achieve social inclusion by closely considering power [41,42] and have relied on Foucault for guidance across diverse conditions [43,44,45,46,47,48,49,50,51].

According to Miller [23], Foucault contributes to thinking about leisure in at least two ways, namely (a) as it connects to individuals’ ability to shape their identity and engage with one another, and (b) as it relates to ways in which institutions use power to perpetuate and promote specific discursive frameworks. First, scholars understand how people become unique and create distinctive identities through discourse and institutions connected to leisure. These identities, when connected to leisure, significantly shape people’s views of and attitudes towards leisure, as well opportunities to experience it. As Clarke and Critcher reasoned, “The meanings expressed through leisure may be ‘received’ from the way society is organized but they can be constructed into unique patterns, given particular slants, even on occasions usurped by the activation of meanings other than those which are socially ‘approved’” [52] (pp. 226–227). Thornton [53] and Blackshaw [54] showed that, through leisure, youth construct their identities around difference, vitality, and solidarity. These analyses help to explain how social networks and associated sub-cultures that establish hierarchies distinguishing members from nonmembers execute power by fostering or preventing leisure. Leisure is facilitated for those included as members of the constructed sub-culture considered to be authentic and, thus, obtain culture capital [55] versus those whose leisure is inhibited since they are identified as mainstream and not members of the sub-culture.

Further connecting identity to power, Miller argues that leisure often relates to demonstrating social class and, thus, reflects and creates power differences. Reissman notes that framing of leisure by people who enjoy high levels of privilege and their creation of practices and institutions around leisure perpetuate their privileges since such engagement reinforces their ability “to dominate the organizational activity, the intellectual life, and the leadership of the community.” [56] (p. 83) People who often experience multiple privileges maintain superiority by accessing economic resources and acquiring cultural capital reflected in activities pursued [57]. Activities such as ballet, equestrian sports, and opera represent higher status. They are often presented in an unwelcoming manner to people experiencing oppression. This contrasts with activities associated with lower status, such as square dancing, county fairs, and country music. Bennett et al. [58] found that, though certain leisure preferences such as music and visual arts showed distinctions between social classes, many variables influence leisure. Markula and Pringle [59] describe how systems of thought exert power and clarify within a culture what is proper relative to characteristics such as gender and nationality. Based on characteristics, including sexual orientation, age, and ability, people treat others as members of categories, thus expecting them to behave in particular ways; often such restrictive thinking and associated expectations occur within the context of leisure, such as within the arts and sports. For example, after studying racism in amateur football in the United Kingdom and Europe, Bradburry [60] recognized that sports reflect historically engrained racialized power relationships. U.S. football also contains cues of gender bias as well [61]. Similarly, Blackshaw [54] identified proliferation of domination via soccer leagues in suburbs distinct from urban games such as basketball.

Conversely, as those encountering oppression engage in some experiences associated with leisure, they can resist the status quo and identify often-hidden privilege. Green argued that “leisure contexts, particularly those with other women, are important spaces for women to review their lives; assessing the balance of satisfactions and activities through contradictory discourses which involve both the ‘mirroring’ of similarities, and resistance to traditional feminine identities.” [62] (p. 171). For example, Radway [63] described how women resisted patriarchal dominance as they collectively identified times for leisure when they freely chose to read romance novels.

The second way Miller [23] identified Foucault’s contribution to leisure studies is by centering on leisure governance, that is, governments’ and institutions’ use of leisure to protect and perpetuate specific discursive frameworks. For instance, heads of state such as U.S. Presidents Kennedy, Reagan, Nixon, and Obama have promoted engagement in physically active recreation to keep populations healthy and strong [64]. Similarly, in the 19th century, factory owners promoted organized recreation activities to control their employees’ lives and promote habits to keep them healthy and fit to work [65]. Numerous scholars (e.g., [66,67,68]) have examined use of withholding opportunities to experience leisure as a method to discipline individuals and ways in which privileged population groups have shaped leisure discourse and mobilized institutions to dominate those targeted to be disadvantaged.

## 4. Power in Inclusive Leisure

Since “power is the problem that has to be resolved” [69] (p. 104), many attempts to promote inclusion are unsuccessful because they fail to consider power adequately. To increase understanding of power’s influence, Foucault commented: “When we examine how, in the late eighteenth century, it was decided to choose imprisonment as the essential mode of punishment, one sees that it was after a long elaboration of various techniques that made it possible to locate people, to fix them in precise places, to constrict them to a certain number of gestures and habits—in short, it was a form of ‘dressage’” [39] (pp. 104–105). Building on Foucault’s insights on power, scholars have proposed various strategies to reduce exclusion and achieve inclusion of all people into leisure experiences that cluster into three approaches.

First, inclusion consists of social considerations associated with oppression and marginalization common across people [70], to create welcoming social and physical contexts in which all participants have opportunities to participate as they make choices and experience a sense of belonging [71]. Centering on conditions associated with oppression is crucial to understand how individuals who experience privileges dominate other people. Concerning building more inclusive contexts, this strategy helps to eliminate differences among people engaging in leisure and fosters development of shared identities, reducing opportunities for individuals to appeal to identity-based differences to exert power over others. As an illustration, although the arts promote creativity and a sense of freedom, people in power control consumption of fine arts, with such consumptive domination reflecting conspicuous leisure [72]. For example, since European nobility engaged in ballroom dance, this activity typically serves as the standard for leisure expression. Research examining ballroom dance reveals that a racially diverse group of dancers perceived that ballroom dance indicates primacy of whiteness [73]. An inclusive leisure orientation rejects the primacy of whiteness by communicating acceptance of diverse forms of dance via opportunities to engage in a range of dances, including Latin forms of ballroom dance.

Second, inclusive leisure involves making accommodations to address individuals’ unique needs so they experience leisure associated with freedom and satisfaction, acquire skills related to leisure, and feel social acceptance [74]. These strategies aim to empower specific individuals to enter leisure contexts and be capable of exerting influence to shape them. Thus, making accommodations helps individuals exercise power against dominant individuals and narratives, as well as increase their capacity to transform discourse. For example, Seattle Parks and Recreation engages those identifying as LGBTQ+ in the planning of inclusive policies and developing accommodations supporting people who historically experience oppression [75]. Furthermore, to accommodate people who experience economic challenges and those experiencing social isolation, the San Francisco Recreation and Park Department converted an undeveloped space into a 60-plot universally designed community garden and gathering space for people who are disenfranchised [76]. In support of such an endeavor, Shinew et al. [77] identified 200 urban community gardeners who reported that the experience helped them to feel connected to their community. Since using a social justice lens encourages the consideration of dynamics of power, focusing on leisure as a context for social justice is one way to promote inclusive leisure [78].

Third, inclusive leisure services aim to unite people around a specific narrative of leisure to exert power collectively. For instance, Gitlin [9] postulates that oppressed members of a culture construct counter-narratives to support tolerance and respect of difference, resulting in weakening of injustices emanating from the dominant culture. From this perspective, instead of studying leisure discourses constructed by the dominant culture to gain and maintain privilege, scholars should center on counter-discourses to deconstruct and present alternatives to the dominant ones [79]. For example, scholars often associate leisure with free time, e.g., [80]; yet this conceptualization of leisure as free time has been extensively critiqued [78,81,82,83], with the continued juxtaposition of leisure and free time reinforcing discourses of privilege, limiting leisure to those privileged enough to have free time [84]. See Figure 1 for a summary of the three approaches to inclusive leisure services that consider influences of power.

The three strategies connect to different modalities of power, specifically power-over, power-to, and power-with. Allen [21] offers an analysis of these modalities to unveil the complex nature of power and elucidate ways in which individuals relate to power. Thus, drawing on her analysis of power is beneficial to understand ways in which inclusive leisure strategies must account for power relations in order to be effective. Allen explains that these modalities are not different forms of power; instead, they “represent analytically distinguishable features of a situation” [21] (p. 37).

## 5. How Does Power-Over Create Barriers to Leisure?

Allen defines power-over “as the ability of an actor or set of actors to constrain the choices available to another actor or set of actors in a nontrivial way” [21] (p. 33). The “non-trivial way” clause is essential in the formulation of this modality of power. It connects limitation of action to options that influence the well-being of individuals whose action is limited. Spracklen [85] identifies that many factors, including gender, nationality, sexuality, and race are entrenched in class domination histories and influence opportunities and constraints to freedom and agency connected to leisure. Hanna et al. [86] recognize if people initiate a constructive difference, power makes it challenging to continue these efforts since there are strong patterns of power that exclude many and favor others.

Two ways in which power structures influence opportunities to experience leisure are consumerism and using leisure as cultural capital. Economic, social, and cultural foci on materialism creates struggles and inequities in life chances, influencing leisure. Rojek [42] notes leisure situates within scarcity, controlled and regulated in a predatory manner by those controlling access and distribution of resources. “Inequalities in access to economic resources and differences in prestige directly influence leisure forms and practices” [42] (p. 149). As leisure connects to capitalism, there is a push toward consumerism [87] with relentless preoccupation to acquire and accumulate goods and services as people receive messages to work more and make more money to purchase more [88]. According to Stebbins [88], people believe continuing to acquire more and better goods contributes to a higher quality of life and moves them closer to perfection and even happiness. For Blackshaw [89], in modernity no leisure pursuits are devoid of connections to consumerism, with part of the allure of consuming being how it fulfills desires immediately. As Cook [90] explains, consuming leisure is integral to enjoyment that could not occur otherwise; however, consumerism is problematic when it moves people away from experiencing leisure that brings meaning and fulfillment and toward leisure as a means to purchase things. For many people in modernity, the meaning of leisure lies in the pleasure of consuming; yet perhaps it is not consuming that people are after, rather it is pursuing happiness [54].

Veblen [91] described the drive to emulate people who have high social status and avoid those judged as inferior; to achieve high status, people need to acquire wealth and display wealth by engaging in conspicuous leisure or conspicuous consumption (accumulating goods and services displayed to demonstrate social position). By focusing on consumptive behaviors, leisure becomes a means for status rather than an end to savor, preventing people from flourishing [92]. Obsession with consumption results in Wang asking, “Why is leisure linked to the market rather than the utopia of free time?” [93] (p. 79). Bourdieu [30] advises that in consumer-based societies, person differentiation occurs most profoundly in how wealth is spent, with leisure becoming a venue to spend money conspicuously to reinforce social standing. Interpreting Bourdieu [55], Featherstone [92] described differential unconscious dispositions and preferences for groups that result in what appear as natural dispositions to engage in certain practices known as taste differences. According to Bourdieu, taste marks social class, identifying leisure practices indicative of particular groups. For instance, some people tend to engage in sports requiring investment of time and money practiced in exclusive places throughout life, such as golf, tennis, sailing, and horseback riding; in comparison, sport participation dramatically declines with age and watching sports increases for people encountering ongoing oppression [92]. As another example, historic disparities in funding for facilities located in certain communities often results in disrepair and neglected leisure spaces [94]. For instance, more than half of the Chicago park budget from 2011 to 2014 was spent on improvements to only 10 of 77 neighborhoods, with seven of the 10 located in mostly white, affluent communities [95].

Often people miss opportunities for leisure when they display what they consume to emulate and advertise their social position [96]. According to Dimanche and Samdahl [96], certain leisure pursuits signal a particular social status. Though people exert, with actions, power over one another, some influence engagement in leisure pursuits to a greater degree, such as parents and peers who often play crucial roles in determining activities in which youth engage [97]. People also base acceptance of other’s behaviors using their preferences; that is, if people’s leisure preferences are different from theirs, they tend to view those selections negatively, thus promoting social inequality [98]. These differences are key in revealing power differentials and identify a need for people experiencing oppression to understand, interpret, and resist domination [99].

## 6. What Are Ways to Promote Power-To that Facilitate Access to Leisure?

Power-to, according to Allen, is “the ability of an individual actor to attain an end or series of ends” [21] (p. 34). Empowerment and resistance are instances of this modality of power. Leisure can be a context to resist power when it creates opportunities for those experiencing oppression to express themselves meaningfully and exert power against individuals who enjoy greater levels of privilege [100]. When accepted to a leisure community, people have a greater capacity to shape dominant discourse and resist domination [101]. As people resist power forces designed to push them down or toward the margins, possibilities for leisure increase and people become empowered [102]. Thus, leisure contexts provide opportunities for various people encountering oppression to develop their capacities and achieve their ends. For instance, youth have used skateboarding to resist capitalistic practices [103]. Similarly, specific social groups use arts as a form of resistance. In particular, youth may use art to express defiance of the dominant narrative organized by adults who enjoy higher opportunities to shape leisure discourse, thus creating counter-narratives. As an example, in the 1930s, German youth used swing dancing to act against restrictions imposed by Nazis [104].

Rojek [42] argues that pluralist societies often think leisure consists of fulfilling as many desires as possible in the easiest and fastest way, bending society to their needs and desires using technical means and efficiency calculation to determine instrumentality. The concern with “an attempt to rationalize leisure for productive means,” is that people evaluate in terms of efficiency [105] (p. 158). Emphasis on instrumental thinking and, as Goldman and Wilson [105] identify, “technical efficiency” prevailing in modernity moves people away from embracing leisure as an end with its experiences of meaning, joy, and happiness, which many scholars (e.g., Aristotle, Pieper) regard as key defining aspects of leisure. As society embraces instrumentality and utilitarianism over enjoyment and fulfillment, people may begin to feel that they lack the capacity to exert any power [106]. Banks [107] argues, while “leisure is offered up as a means of free and autonomous expression, it may be leading, paradoxically, to the erosion of freedom as the terrain of critical and disinterested leisure is pervasively colonized by discourses of economic rationality” [108] (p. 668).

To address emphasis on instrumentalism and efficiency resulting in a dichotomy between work and leisure in modernity, Beatty and Torbet [81] offer a metaphor of the Yin-Yang symbol to illustrate how leisure intertwines with work, with boundaries often blurred. Leisure is “... conceptually distinguished from free time (activities). Leisure is rather the structuring principle of ethic underneath free time (activities), since its scope is broader than activities and time alone. It is about becoming fully human” [109] (p. 112). From this perspective, leisure is not instrumental but rather ethical, because it contributes to human development and fulfillment [110]. Expanding this “humanistic narrative,” Kjølsrød [83] proposes leisure must be authentic, compatible with ethical behavior, and a source of knowledge and social resilience.

## 7. How Do People Collaborate to Exert Power-With?

In Allen’s view, power-with is “the human ability not just to act but to act in concert” [21] (p. 35). Thus, this modality of power has to do with people’s capacity to work together for a shared goal or purpose against a system of domination; that is, solidarity. Leisure spaces can create opportunities for people to come together in a shared commitment. For instance, Richardson [111] identified the conservation organization, Outdoor Afro, a network celebrating and inspiring people of color to connect with nature. In addition, Kristiansen et al. [112] interviewed long-term volunteers about their commitment to maintaining the ski-flying hill in Vikersund, Norway, identifying that actions of the volunteers contributed to a collective identity around the ski-flying hill. This maintained and reinforced a strong community identity and social solidarity by achieving local resistance to perceived hostility from outside organizations.

Similarly, Wigginton [113] identifies sport as a resource of hope for many who are traditionally oppressed. In another example, since cricket has a strong connection to British imperialism, as Britain forced conquered nations to adopt British leisure preferences, some players used cricket to express solidarity and cultural pride by being victorious on the field [114]. Engagement in the arts can cultivate a sense of solidarity as people experience power-with. For example, Suriano identified that artistic collaboration, such as liberation songs, with South Africans amplified and solidified the ability of Tanzanians to transcend nationalism and shaped “convivial transnational solidarity” [115] (p. 985). Further, Webb and Webb-Gannon examined lyrical, musical, and visual devices of popular Melanesian songs and accompanying videos that promote regional identity and solidarity in Brisbane, Australia and found they contributed to “the idea of one skin or blackness as distinctive, thus turning the pejorative associations and experiences of being labeled the black nesia into a feature to celebrate” [116] (p. 59).

The trust people place in others promotes a sense of solidarity and correlates strongly with power [117]. There is a robust relationship between trust and social inclusion within leisure [118]. For example, to develop trust between participants and professionals, renovations to Brooklyn’s Prospect Park began by working with people residing in surrounding communities, including many immigrants, to determine desirable spaces [119]. In another example, Collum [120] identified a way to develop trust and connect with community members not proficient at English by using a mobile recreation team in Boise, Idaho to establish temporary play spaces serving a large refugee population. Power noted trust is key in creating a sense of belonging for people encountering oppression that often “requires considerable time and energy” to create [121] (p. 74). See Figure 2 for a summary of how power-to and power-with can help people become more fully human through ethical leisure.

## 8. Conclusions

When people benefit from oppression of others, often they support continuing systems that promote power differences. Because of privilege, people often are unaware or do not believe they contribute to marginalization and are not fully cognizant of unjust situations of people experiencing domination. Everyone can exert some level of power on the way communities shape leisure and how individuals engage in and experience it. However, since leisure professionals occupy privileged positions in leisure communities, they have numerous opportunities to advocate for equity and inclusion for all. Because achieving this goal can be difficult, considering how power influences people’s leisure, we have identified actions to promote inclusive leisure. Everyone experiences some level of domination and can recall feelings of powerlessness. Yet, people also have possibilities to exert power. In particular, we have examined how people can exert power-over, power-to, and power-with to shape leisure and promote inclusive leisure.

Too many people encounter regular, systematic, and far-reaching deprivation of power. Sometimes, people who have limited encounters with oppression use such experiences to gain insight and perspective into the lives of those who experience oppression regularly and that increases a willingness to provide inclusive leisure. Unfortunately, for some, these few experiences provide a bitter taste of repression, motivating them to marginalize others, supporting the destructive, negative belief that by oppressing others they maintain and, even, expand their power. What we believe is less important is where each person fits into the continuum of power; what is more important is how people reflect on their challenges and use their power to create equitable and inclusive leisure.

Often, people’s power-to hardly suffices to combat situations where they encounter exploitation and oppression. These individuals may lack power to alter such a situation due to other agents’ power-over (or systemic power-over), restricting individuals’ ability to achieve their goals to promote equitable and inclusive leisure. For instance, one might be born into a family or society that discourages engagement in particular leisure experiences (see [122]). When individuals lack power-to, in some cases, power-with becomes the only way for them to promote inclusive leisure goals, as we have shown in Section 7. For example, to develop trust between community participants, especially those who recently immigrated from another country, and leisure service professionals, renovations to Brooklyn’s Prospect Park began by park personnel working with people residing in surrounding communities, including many immigrants, to determine desirable spaces [119]. In other cases, the power exerted over them is so strong that they will be unable to promote inclusive leisure. As an illustration, because women are identified as the primary caretakers of children in our society, when compared to men they are often more constrained in their leisure because of the gender role society typically assigns to them [123]. Yet, reflecting on these forms of power and understanding how they connect to leisure experiences, as well as to how individuals make sense of leisure, helps to unveil the systemic forces (or discursive elements) that exert power over them. This understanding, as Foucault reasons in his description of philosophy at the beginning of this work, is crucial to eliminate situations of oppression [124].

## Figures and Tables

**Figure 1 ijerph-18-02220-f001:**
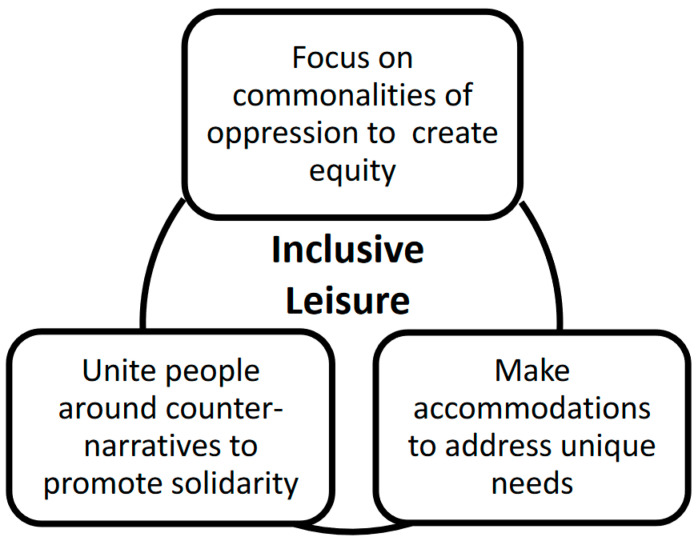
Approaches to inclusive leisure that consider influences of power.

**Figure 2 ijerph-18-02220-f002:**
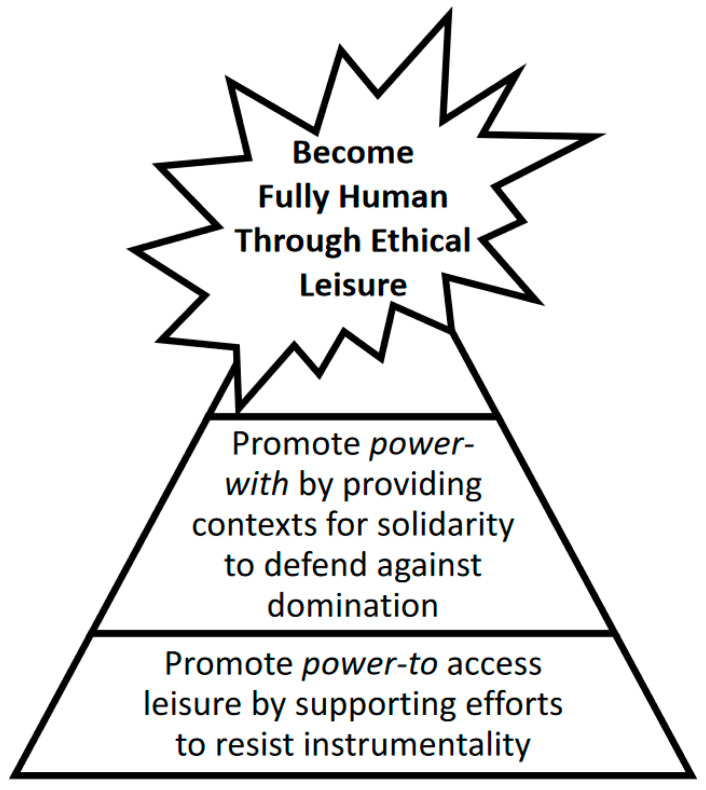
Contributions of power-to and power-with associated with experiencing leisure.

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
