# Peer review of "The Influence of Power on Leisure: Implications for Inclusive Leisure Services"

_ijerph, 2021, doi:10.3390/ijerph18052220_

Round 1

Reviewer 1 Report

This is an insightful manuscript. I appreciate the opportunity to review this manuscript.  

The authors have not fully incorporated the template, and some excessive text from the template still remains. Please delete text from the template under 1. Introduction (line 69).  

line 297: The "two ways" are not explicitly mentioned. It would be easier for the readers to identify which "two ways" you refer to. 

Although the authors provide expectations to the readers that the paper focuses on Foucault, sections 5~8 did not really give an impression that the authors really did so.   

Author Response

This is an insightful manuscript. I appreciate the opportunity to review this manuscript.  

The authors have not fully incorporated the template, and some excessive text from the template still remains. Please delete text from the template under 1. Introduction (line 69).  

Thanks for pointing that out.

line 297: The "two ways" are not explicitly mentioned. It would be easier for the readers to identify which "two ways" you refer to. 

see lines 164-167. 

Although the authors provide expectations to the readers that the paper focuses on Foucault, sections 5~8 did not really give an impression that the authors really did so.   

We use these sections to illustrate Allen’s distinction among power-over, power-to, and power-with. These forms of power are not strictly speaking Foucault’s. Yet, Allen derives them from her reading of Foucaldian scholars.  

Reviewer 2 Report

The article raises a topic of great interest - the issue of power in leisure - scarcely visible in the corpus of scientific knowledge on this topic. For this reason, the contribution is very suitable for the monograph for which it has been proposed.

However, some form and content changes are required so that the article fits into the parameters of excellence required by the journal.

Regarding questions of form:

  • The text between lines 70-78 should be removed as it corresponds to the text in the Journal template. Instead, the paragraphs that, in the current version, now appear on lines 20 to 28 should be placed.
  • In order to maintain the parallel structure of the academic writing of the manuscript, it is suggested to delete the title of the work "The Rise of Creative Class" (line 374), since no other titles are mentioned throughout the manuscript.

Regarding content aspects:

  • Foucault's quote (lines 141-144) is misleading as written. Since the original fragment refers to sexuality, it must be specified that this quote is useful with respect to leisure, so it would be appropriate to place, in square brackets, the word "leisure" even though it is not the object of study to which the original text refers .
  • The characterization and analysis of leisure activities in the 19th century can be enriched by the analysis of Codina (1990: http://diposit.ub.edu/dspace/handle/2445/42752).
  • When considering cultural capital in relation to leisure, it is suggested to incorporate the case in which a person is socialized in the practice of certain leisure activity as a result of a family tradition, given that it is an important factor in the experience of the aforesaid leisure activity (https://www.tandfonline.com/doi/abs/10.1080/14613808.2013.859660).
  • Finally, the article must thoroughly review the conclusions it reaches. Certainly, given that the article is written by two leisure scholars from the Anglo-Saxon context, it makes the predominant perspective corresponding to this position and the power it carries with. However, statements such as "What we believe is important is how each person reflects on their challenges and uses their power to create equitable and inclusive leisure" (lines 444-446) threaten the argumentative solidity of the text: Should responsibility be individual the final message of a social problem? This ending is reminiscent of the problems of self-help, which lead to the person - extolling him - what is the inability of the system to take charge of the individuals with whom it has carried out its social contract.
  • Resolving this contradiction is not easy, since, in an extreme case, it would imply denying oneself as a member of a certain social group (with specific privileges and oppressions). However, cases of self-managed leisure or that leisure practices contravene the law (some cases Rojek has pointed out) can be introduced in the text as examples of emancipation outside the margins - and, decidedly, forming part of social deviation. Another possible reflection is that of the very marginalization of leisure scholars before the leisure industry or the corresponding government policies: here is a flank towards which to direct emancipatory actions from the universities.

Reviewing an article also puts reviewers in a position of power. In this case, an attempt has been made to offer - constructively - suggestions for one's own emancipation from the limitations of the academic world itself.

With the proposed modifications and the resolution of its discursive contradictions, the article will brilliantly contribute to the theme "Leisure and Time Management in Fostering Wellbeing and Health: Current Issues and New Trends" of the IJERPH monograph.

Author Response

The article raises a topic of great interest - the issue of power in leisure - scarcely visible in the corpus of scientific knowledge on this topic. For this reason, the contribution is very suitable for the monograph for which it has been proposed.

However, some form and content changes are required so that the article fits into the parameters of excellence required by the journal.

Regarding questions of form:

  • The text between lines 70-78 should be removed as it corresponds to the text in the Journal template. Instead, the paragraphs that, in the current version, now appear on lines 20 to 28 should be placed.

Thanks for pointing that out.

  • In order to maintain the parallel structure of the academic writing of the manuscript, it is suggested to delete the title of the work "The Rise of Creative Class" (line 374), since no other titles are mentioned throughout the manuscript.

Thanks for pointing that out.

Regarding content aspects:

  • Foucault's quote (lines 141-144) is misleading as written. Since the original fragment refers to sexuality, it must be specified that this quote is useful with respect to leisure, so it would be appropriate to place, in square brackets, the word "leisure" even though it is not the object of study to which the original text refers .

Noted. We’ve removed the word “leisure” and specified that Foucault’s analysis is in the context of his understanding of “sex” as a discursive object. Thus, his method can be used to approach other discursive objects such as leisure.

  • The characterization and analysis of leisure activities in the 19th century can be enriched by the analysis of Codina (1990: http://diposit.ub.edu/dspace/handle/2445/42752).
  • When considering cultural capital in relation to leisure, it is suggested to incorporate the case in which a person is socialized in the practice of certain leisure activity as a result of a family tradition, given that it is an important factor in the experience of the aforesaid leisure activity (https://www.tandfonline.com/doi/abs/10.1080/14613808.2013.859660).

See lines 466-467 and reference 123

  • Finally, the article must thoroughly review the conclusions it reaches. Certainly, given that the article is written by two leisure scholars from the Anglo-Saxon context, it makes the predominant perspective corresponding to this position and the power it carries with.

We have included a paragraph acknowledging our position within the academic field of leisure studies. See it in lines xxx- xxx. Thanks for pointing this out.

  • However, statements such as "What we believe is important is how each person reflects on their challenges and uses their power to create equitable and inclusive leisure" (lines 444-446) threaten the argumentative solidity of the text: Should responsibility be individual the final message of a social problem? This ending is reminiscent of the problems of self-help, which lead to the person - extolling him - what is the inability of the system to take charge of the individuals with whom it has carried out its social contract.

We have added one paragraph at the end of the paper addressing this concern and highlighting the value of understanding the three forms of power outlined in the paper.

  • Resolving this contradiction is not easy, since, in an extreme case, it would imply denying oneself as a member of a certain social group (with specific privileges and oppressions). However, cases of self-managed leisure or that leisure practices contravene the law (some cases Rojek has pointed out) can be introduced in the text as examples of emancipation outside the margins - and, decidedly, forming part of social deviation. Another possible reflection is that of the very marginalization of leisure scholars before the leisure industry or the corresponding government policies: here is a flank towards which to direct emancipatory actions from the universities.

We have touched on this difficulty at the end of the last paragraph by mentioning the importance of power-with and situations in which individuals are powerless and can hardly pursue their goals (exert power-to). We’ve also used an example to illustrate it. 

Reviewing an article also puts reviewers in a position of power. In this case, an attempt has been made to offer - constructively - suggestions for one's own emancipation from the limitations of the academic world itself.

With the proposed modifications and the resolution of its discursive contradictions, the article will brilliantly contribute to the theme "Leisure and Time Management in Fostering Wellbeing and Health: Current Issues and New Trends" of the IJERPH monograph.

Reviewer 3 Report

The Influence of Power on Leisure: Implications for Inclusive Leisure Services

Introduction and theoretical background

  • The research purpose and necessity of the introduction section should be described in more detail. For instance, with regard to the research necessity, a specific description of inclusive leisure should be presented to clarify research problems.
  • Also, specific contents related to leisure experience opportunity should be described, based on relevant previous studies.
  • In addition, feminist theory-based research problems should be described in the introduction section, on the basis of relevant previous studies.

Research method and procedure

- The text has to offer a definite explanation of literature research method and procedure that are applied to this study.

  • Also, there should be a description of how reliability and validity of this study content were acquired.

Research result and discussion

  • It is required to clearly classify the novel findings that this study suggests. Moreover, a meaningful discussion should be described through comparison between the result of this study and those of the existing studies.
  • Also, the three criteria(L-153, L-221, L-286) specified in the result section do not correspond to the necessity and purpose of this study.
  • The conclusion of this study should show a meaningful depiction of what the result of this study means.

Overall review

  • On the whole, it is hard to figure out the meaning of the study(l-430). Besides, the text has a lot of results that have already been revealed in the existing studies, and they are abstract. Accordingly, a lot of revisions are required.

Author Response

Introduction and theoretical background

  • The research purpose and necessity of the introduction section should be described in more detail. For instance, with regard to the research necessity, a specific description of inclusive leisure should be presented to clarify research problems.

We have added a paragraph specifying the nature of philosophical analysis and its contribution to leisure studies.

  • Also, specific contents related to leisure experience opportunity should be described, based on relevant previous studies.
  • In addition, feminist theory-based research problems should be described in the introduction section, on the basis of relevant previous studies.

We don’t think the reference to the context in which Allen formulates her analyses of power forces us to get into that literature. Indeed, throughout the paper, we defend the transcontextual character of such analyses as well as the notion of power. Foucault didn’t formulate his examination of power by analyzing leisure experiences. Yet, we can use his insights to better understand discursive aspects of leisure without immersing ourselves in the original contexts in which Foucault developed his studies (e.g., sex, mental health institutions, and prisons).

Research method and procedure

- The text has to offer a definite explanation of literature research method and procedure that are applied to this study.

  • Also, there should be a description of how reliability and validity of this study content were acquired.

 We have added a paragraph specifying the nature of philosophical analysis and its contribution to leisure studies.

Research result and discussion

  • It is required to clearly classify the novel findings that this study suggests. Moreover, a meaningful discussion should be described through comparison between the result of this study and those of the existing studies.
  • Also, the three criteria(L-153, L-221, L-286) specified in the result section do not correspond to the necessity and purpose of this study.

We have added a paragraph specifying the nature of philosophical analysis and its contribution to leisure studies.

  • The conclusion of this study should show a meaningful depiction of what the result of this study means.

We have added a paragraph to the conclusion, clarifying the relevance of our analysis and providing an example.

Overall review

  • On the whole, it is hard to figure out the meaning of the study(l-430). Besides, the text has a lot of results that have already been revealed in the existing studies, and they are abstract. Accordingly, a lot of revisions are required.

Reviewer 4 Report

REVIEW

THE INFLUENCE OF POWER ON LEISURE: IMPLICATIONS FOR INCLUSIVE 2 LEISURE SERVICES

REVIEWER'S REPORT

The article meets the stated objective, is well structured, contributes to the field of leisure and psychology by relating leisure and self-esteem, and presents recent findings in the systematic review they have conducted. In general terms I liked the way it is written in structure, content and format.

In the following, I detail in each section some positive considerations and suggestions for minor changes.

SUMMARY

It is correct

KEY WORDS

Are correct

  1. INTRODUCTION

From line 20 to line 68 I consider it to be the introduction of the article.

The paragraph that seems properly in the introduction does not provide content but general guidelines from line 70 to line 78.

Once explained the context or the priority focus on inclusion issues, it leads us to see it from Foucault's point of view.

  1. THE OMINPRESENCE OF POWER

This section presents the content in an orderly, well-argued and adequately referenced manner.

  1. THE POWER OF LEISURE

As in the previous section, the content is orderly, well argued and adequately referenced.

  1. THE POWER OF INCLUSIVE LEISURE

As in sections 2 and 3, this one is orderly, well argued and adequately referenced. In addition to being one of the core themes of the article.

  1. HOW DOES POWER-OVER CREATE BARRIERS TO LEISURE?

  The content is very interesting and requires researchers to reflect on the relationship between power and real opportunities for leisure experiences.

  1. WHAT ARE WAYS TO PROMOTE POWER-TO ACCESS LEISURE?

  In this part it is very interesting and novel to change the focus and to resort to the importance of power in order to promote inclusion in leisure.

  1. HOW DO PEOPLE COLLABORATE TO EXERT POWER-WITH?

I found the arguments and their corresponding references very interesting.

  1. CONCLUSIONS

I consider it appropriate to specify the conclusions a little more and therefore to expand this section.

REFERENCES

The references used are abundant, pertinent, of high quality, in journals of high impact and closely related to the object of study.

Author Response

Thanks for your comments on the sections. We’ve removed the intro paragraph from the template and emphasized some of the messages that you highlight in your comments on the sections.

Reviewer 5 Report

This paper is a study that analyzes  how  discourse influences leisure and identify ways to facilitate inclusive leisure. The Authors concluded that each person can make positive contributions and offer inclusive leisure. They used a Foucauldian perspective to analyse a phenomen that influences people’s opportunities to experience leisure.

The presented manuscript is 12 pages long. The theoretical material has not been illustrated in any schemes or figures. This is the weakness of this manuscript. Graphic presentations of the theoretical frame would certainly improve the quality of the work and help to interpret the final conclusions. The adopted formal structure do not meet the requirements of the editors of the Journal and the Special Issue "Leisure and Time Management in Fostering Wellbeing and Health: Current Issues and New Trends. There are ambiguities that require some clarification:

  1. The text of the manuscript should consists of the following sections: Introduction, Materials and Methods, Results, Discussion, Conclusions. The structure of this manuscript is different, it is prepared from the editorial side, not very carefully.
  2. The whole Introduction (lines:69-78) section includes information from Instructions for Authors about manuscript preparation.
  3. In the Conclusions section there are no specific indications and practical conclusions, is written too generally. The manuscript is typically theoretical in nature. The Authors conducted a thorough review of the subject literature. References to practice are lacking. The Authors did not relate their recommendations to a specific social setting and social and national background, which seems misguided.

Overall, the content of this research is not only cognitively interesting but also encourages further discussion and research. I hope that the suggestions discussed above will be used to improve the quality of this paper.

Author Response

This paper is a study that analyzes  how  discourse influences leisure and identify ways to facilitate inclusive leisure. The Authors concluded that each person can make positive contributions and offer inclusive leisure. They used a Foucauldian perspective to analyse a phenomen that influences people’s opportunities to experience leisure.

The presented manuscript is 12 pages long. The theoretical material has not been illustrated in any schemes or figures. This is the weakness of this manuscript. Graphic presentations of the theoretical frame would certainly improve the quality of the work and help to interpret the final conclusions. The adopted formal structure do not meet the requirements of the editors of the Journal and the Special Issue "Leisure and Time Management in Fostering Wellbeing and Health: Current Issues and New Trends. There are ambiguities that require some clarification:

We’ve added two figures

  1. The text of the manuscript should consists of the following sections: Introduction, Materials and Methods, Results, Discussion, Conclusions. The structure of this manuscript is different, it is prepared from the editorial side, not very carefully.
  2. The whole Introduction (lines:69-78) section includes information from Instructions for Authors about manuscript preparation.
  3. In the Conclusions section there are no specific indications and practical conclusions, is written too generally. The manuscript is typically theoretical in nature. The Authors conducted a thorough review of the subject literature. References to practice are lacking. The Authors did not relate their recommendations to a specific social setting and social and national background, which seems misguided.

Overall, the content of this research is not only cognitively interesting but also encourages further discussion and research. I hope that the suggestions discussed above will be used to improve the quality of this paper.

In the introduction, we have added a paragraph specifying the nature of philosophical analysis and its contribution to leisure studies. In the conclusion, we have entered a paragraph, clarifying the relevance of our analysis and providing an example.

Round 2

Reviewer 5 Report

After having verified the response to the review of the manuscript “The Influence of Power on Leisure: Implications for Inclusive  Leisure Services” and to the revised version of article, I would like to confirm that the current form of the manuscript almost meets the requirements of  publication in the IJERPH journal. The authors have taken into account the major part of the reviewer’s comments.